# VARIABLE COMPUTATION IN RECURRENT NEURAL NETWORKS

**Yacine Jernite** [*]
Department of Computer Science
New York University
New York, NY 10012, USA
`jernite@cs.nyu.edu`

**Edouard Grave, Armand Joulin & Tomas Mikolov**
Facebook AI Research
New York, NY 10003, USA
`{egrave,ajoulin,tmikolov}@fb.com`

## ABSTRACT

Recurrent neural networks (RNNs) have been used extensively and with increasing success to model various types of sequential data. Much of this progress has been achieved through devising recurrent units and architectures with the flexibility to capture complex statistics in the data, such as long range dependency or localized attention phenomena. However, while many sequential data (such as video, speech or language) can have highly variable information flow, most recurrent models still consume input features at a constant rate and perform a constant number of computations per time step, which can be detrimental to both speed and model capacity. In this paper, we explore a modification to existing recurrent units which allows them to learn to vary the amount of computation they perform at each step, without prior knowledge of the sequence's time structure. We show experimentally that not only do our models require fewer operations, they also lead to better performance overall on evaluation tasks.

## 1 INTRODUCTION

The class of Recurrent Neural Network models (RNNs) is particularly well suited to dealing with sequential data, and has been successfully applied to a diverse array of tasks, such as language modeling and speech recognition (Mikolov, 2012), machine translation (Mikolov, 2012; Cho et al., 2014a), or acoustic modeling (Robinson et al., 1993; Graves & Jaitly, 2014) among others. Two factors have been instrumental in allowing this paradigm to be so widely adopted and give rise to the aforementioned successes. On the one hand, recent advances in both hardware and software have had a significant role in bringing the training of recurrent models to tractable time periods. On the other hand, novel units and architectures have allowed recurrent networks to model certain features of sequential data better than Elman's simple RNN architecture (Elman, 1990). These include such developments as the LSTM (Hochreiter & Schmidhuber, 1997) and GRU (Cho et al., 2014a) units, which can more easily learn to model long range interactions (Chung et al., 2014), or attention mechanisms that allow the model to focus on a specific part of its history when making a prediction (Bahdanau et al., 2014). In this work, we focus on another feature of recurrent networks: the ability to efficiently model processes happening at different and possibly varying time scales.

Most existing recurrent models take one of two approaches regarding the amount of computation they require. Either the computational load is constant over time, or it follows a fixed (or deterministic) schedule (Koutník et al., 2014), (Mikolov et al., 2014). The latter approach has proven especially useful when dealing with sequences which reflect processes taking place at different lev-

---

[*] Work done at Facebook AI Research

els (and time scales) (Bojanowski et al., 2015). However, we believe that taking a more flexible approach could prove useful.

Consider sequential data such as video feeds, audio signal, or language. In video data, there are time periods where the frames differ very slightly, and where the underlying model should probably do much less computation than when the scene completely changes. When modeling speech from an audio signal, it is also reasonable to expect that the model should be able do little to no computation during silences. Finally, in the case of character level language modeling, having more computational power at word boundaries can certainly help: after reading the left context *The prime...*, the model should be able to put a higher likelihood on the sequence of characters that make up the word *minister*. However, we can take this idea one step further: after reading *The prime min...*, the next few characters are almost deterministic, and the model should require little computation to predict the sequence *i-s-t-e-r*.

In this work, we show how to modify two commonly used recurrent unit architectures, namely the Elman and Gated Recurrent Unit, to obtain their variable computation counterparts. This gives rise to two new architecture, the Variable Computation RNN and Variable Computation GRU (VCRNN and VCGRU), which take advantage of these phenomena by deciding at each time step how much computation is required based on the current hidden state and input. We show that the models learn time patterns of interest, can perform fewer operations, and may even take advantage of these time structures to produce better predictions than the constant computation versions.

We start by giving an overview of related work in Section 2, provide background on the class of Recurrent Neural Networks in Section 3, describe our model and learning procedure in Section 4, and present experimental results on music as well as bit and character level language modeling in section 5. Finally, Section 6 concludes and lays out possible directions for future work.

## 2 RELATED WORK

How to properly handle sequences which reflect processes happening at different time scales has been a widely explored question. Among the proposed approaches, a variety of notable systems based on Hidden Markov Models (HMMs) have been put forward in the last two decades. The Factorial HMM model of (Ghahramani & Jordan, 1997) (and its infinite extension in (Gael et al., 2008)) use parallel interacting hidden states to model concurrent processes. While there is no explicit handling of different time scales, the model achieves good held-out likelihood on Bach chorales, which exhibit multi-scale behaviors. The hierarchical HMM model of (Fine et al., 1998) and (Murphy & Paskin, 2001) takes a more direct approach to representing multiple scales of processes. In these works, the higher level HMM can recursively call sub-HMMs to generate short sequences without changing its state, and the authors show a successful application to modeling cursive writing. Finally, the Switching State-Space Model of (Ghahramani & Hinton, 2000) combines HMMs and Linear Dynamical Systems: in this model, the HMM is used to switch between LDS parameters, and the experiments show that the HMM learns higher-level, slower dynamics than the LDS.

On the side of Recurrent Neural Networks, the idea that the models should have mechanisms that allow them to handle processes happening at different time scales is not a new one either. On the one hand, such early works as (Schmidhuber, 1991) and (Schmidhuber, 1992) already presented a two level architecture, with an "automatizer" acting on every time step and a "chunker" which should only be called when the automatizer fails to predict the next item, and which the author hypothesizes learns to model slower scale processes. On the other hand, the model proposed in (Mozer, 1993) has slow-moving units as well as regular ones, where the slowness is defined by a parameter $\tau \in [0, 1]$ deciding how fast the representation changes by taking a convex combination of the previous and predicted hidden state.

Both these notions, along with different approaches to multi-scale sequence modeling, have been developed in more recent work. (Mikolov et al., 2014) expand upon the idea of having slow moving units in an RNN by proposing an extension of the Elman unit which forces parts of the transition matrix to be close to the identity. The idea of having recurrent layers called at different time steps has also recently regained popularity. The Clockwork RNN of (Koutník et al., 2014), for example, has RNN layers called every 1, 2, 4, 8, etc... time steps. The conditional RNN of (Bojanowski et al., 2015) takes another approach by using known temporal structure in the data: in the character level

level language modeling application, the first layer is called for every character, while the second is only called once per word. It should also be noted that state-of-the art results for language models have been obtained using multi-layer RNNs (Józefowicz et al., 2016), where the higher layers can in theory model slower processes. However, introspection in these models is more challenging, and it is difficult to determine whether they are actually exhibiting significant temporal behaviors.

Finally, even more recent efforts have considered using dynamic time schedules. (Chung et al., 2016) presents a multi-layer LSTM, where each layer decides whether or not to activate the next one at every time step. They show that the model is able to learn sensible time behaviors and achieve good perplexity on their chosen tasks. Another implementation of the general concept of adaptive time-dependent computation is presented in (Graves, 2016). In that work, the amount of computation performed at each time step is varied not by calling units in several layers, but rather by having a unique RNN perform more than one update of the hidden state on a single time step. There too, the model can be shown to learn an intuitive time schedule.

In this paper, we present an alternative view of adaptive computation, where a single Variable Computation Unit (VCU) decides dynamically how much of its hidden state needs to change, leading to both savings in the number of operations per time step and the possibility for the higher dimensions of the hidden state to keep longer term memory.

## 3 RECURRENT NEURAL NETWORKS

Let us start by formally defining the class of Recurrent Neural Networks (RNNs). For tasks such as language modeling, we are interested in defining a probability distribution over sequences $\mathbf{w} = (w_1, \ldots, w_T)$. Using the chain rule, the negative log likelihood of a sequence can be written:

$$\mathcal{L}(\mathbf{w}) = -\sum_{t=1}^{T} \log\big(p\big(w_t | \mathcal{F}(w_1, \ldots, w_{t-1})\big)\big).$$  (1)

where $\mathcal{F}$ is a filtration, a function which summarizes all the relevant information from the past. RNNs are a class of models that can read sequences of arbitrary length to provide such a summary in the form of a hidden state $h_t \approx \mathcal{F}(w_1, \ldots, w_{t-1})$, by applying the same operation (recurrent unit) at each time step. More specifically, the recurrent unit is defined by a recurrence function $g$ which takes as input the previous hidden state $h_{t-1}$ at each time step $t$, as well as a representation of the input $x_t$ (where $h_{t-1}$ and $x_t$ are $D$-dimensional vectors), and (with the convention $h_0 = 0$,) outputs the new hidden state:

$$h_t = g(h_{t-1}, x_t)$$  (2)

**Elman Unit.**   The unit described in (Elman, 1990) is often considered to be the standard unit. It is parametrized by $U$ and $V$, which are square, $D$-dimensional transition matrices, and uses a sigmoid non-linearity to obtain the new hidden state:

$$h_t = \tanh(U h_{t-1} + V x_t).$$  (3)

In the Elman unit, the bulk of the computation comes from the matrix multiplications, and the cost per time step is $O(D^2)$. In the following section, we show a simple modification of the unit which allows it to reduce this cost significantly.

**Gated Recurrent Unit.**   The Gated Recurrent Unit (GRU) was introduced in (Cho et al., 2014b). The main difference between the GRU and Elman unit consists in the model's ability to interpolate between a proposed new hidden state and the current one, which makes it easier to model longer range dependencies. More specifically, at each time step $t$, the model computes a reset gate $r_t$, an update gate $z_t$, a proposed new hidden state $\tilde{h}_t$ and a final new hidden state $h_t$ as follows:

$$r_t = \sigma(U_r h_{t-1} + V_r x_t), \qquad z_t = \sigma(U_z h_{t-1} + V_z x_t)$$  (4)

$$\tilde{h}_t = \tanh(U(r_t \odot h_{t-1}) + V x_t)$$  (5)

And:

$$h_t = z_t \odot \tilde{h}_t + (1 - z_t) \odot h_{t-1}$$  (6)

Where $\odot$ denotes the element-wise product.

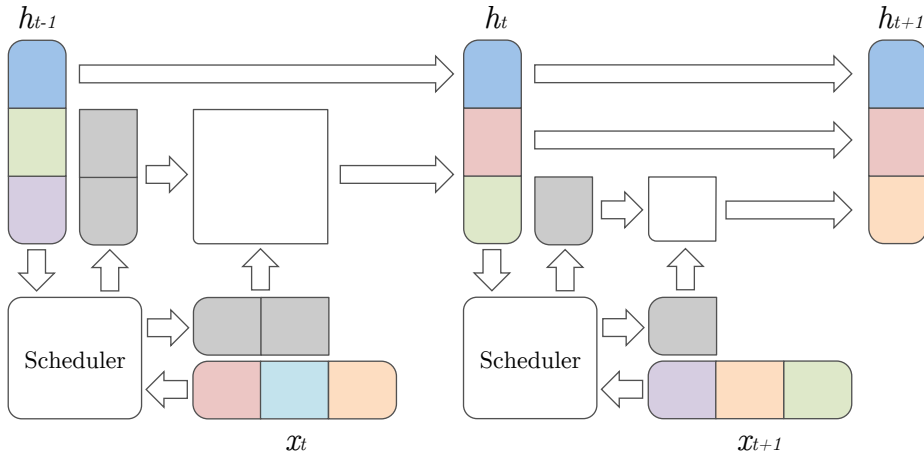

Figure 1: Two time steps of a VCU. At each step $t$, the scheduler takes in the current hidden vector $h_{t-1}$ and input vector $x_t$ and decides on a number of dimensions to use $d$. The unit then uses the first $d$ dimensions of $h_{t-1}$ and $x_t$ to compute the first $d$ elements of the new hidden state $h_t$, and carries the remaining $D - d$ dimensions over from $h_{t-1}$.

## 4 VARIABLE COMPUTATION RNN

As noted in the previous section, the bulk of the computation in the aforementioned settings comes from the linear layers; a natural option to reduce the number of operations would then be to only apply the linear transformations to a sub-set of the hidden dimensions. These could in theory correspond to any sub-set indices in $\{1, \ldots, D\}$; however, we want a setting where the computational cost of the choice is much less than the cost of computing the new hidden state. Thus, we only consider the sets of first $d$ dimensions of $\mathbb{R}^D$, so that there is a single parameter $d$ to compute.

Our Variable Computation Units (VCUs) implement this idea using two modules: a *scheduler* decides how many dimensions need to be updated at the current time step, and the VCU performs a *partial update* of its hidden state accordingly, as illustrated in Figure 1. Section 4.1 formally describes the scheduler and partial update operations, and Section 4.2 outlines the procedure to jointly learn both modules.

### 4.1 MODEL DESCRIPTION

**Scheduler.** The model first needs to decide how much computation is required at the current time step. To make that decision, the recurrent unit has access to the current hidden state and input; this way, the model can learn to ignore an uninformative input, or to decide on more computation when an it is unexpected given the current hidden state. The *scheduler* is then defined as a function $m : \mathbb{R}^{2D} \to [0, 1]$ which decides what portion of the hidden state to change based on the current hidden and input vectors. In this work, we decide to implement it as a simple log-linear function with parameter vectors $u$ and $v$, and bias $b$, and at each time step $t$, we have:

$$m_t = \sigma(u \cdot h_{t-1} + v \cdot x_t + b). \tag{7}$$

**Partial update.** Once the scheduler has decided on a computation budget $m_t$, the VCU needs to perform a *partial update* of the first $\lceil m_t D \rceil$ dimensions of its hidden state. Recall the hidden state $h_{t-1}$ is a $D$-dimensional vector. Given a smaller dimension $d \in \{1, \ldots, D\}$, a partial update of the hidden state would take the following form. Let $g_d$ be the $d$-dimensional version of the model's recurrence function $g$ as defined in Equation 2, which uses the upper left $d$ by $d$ square sub-matrices of the linear transformations $(U_d, V_d, \ldots)$, and $h_{t-1}^d$ and $x_t^d$ denote the first $d$ elements of $h_{t-1}$ and $x_t$. We apply $g_d$ to $h_{t-1}^d$ and $x_t^d$, and carry dimensions $d + 1$ to $D$ from the previous hidden state, so the new hidden state $h_t$ is defined by:

$$h_t^d = g_d(h_{t-1}^d, x_t^d) \quad \text{and} \quad \forall i > d, \ h_{t,i} = h_{t-1,i}.$$

**Soft mask.** In practice, the transition function we just defined would require making a hard choice at each time step of the number of dimensions to be updated, which makes the model non-differentiable and can significantly complicate optimization. Instead, we approximate the hard choice by using a gate function to apply a *soft mask*. Given $m_t \in [0, 1]$ and a sharpness parameter $\lambda$, we use the gating vector $e_t \in \mathbb{R}^D$ defined by:

$$\forall i \in 1, \ldots, D, \ (e_t)_i = \text{Thres}_\epsilon \left( \sigma \big( \lambda(m_t D - i) \big) \right), \tag{8}$$

where $\text{Thres}_\epsilon$ maps all values greater than $1 - \epsilon$ and smaller than $\epsilon$ to 1 and 0 respectively. That way, the model performs an update using the first $(m_t \times D + \eta)$ dimensions of the hidden state, where $\eta$ goes to 0 as $\lambda$ increases, and leaves its last $((1 - m_t) \times D - \eta)$ dimensions unchanged. Thus, if $g$ is the recurrence function defined in Equation 2, we have:

$$\bar{h}_{t-1} = e_t \odot h_{t-1}, \quad \bar{i}_t = e_t \odot x_t, \quad \text{and} \quad h_t = e_t \odot g(\bar{h}_{t-1}, \bar{x}_t) + (1 - e_t) \odot h_t. \tag{9}$$

The computational cost of this model at each step $t$, defined as the number of multiplications involving possibly non-zero elements is then $O(m_t^2 D^2)$. The construction of the VCRNN and VCGRU architectures using this method is described in the appendix.

## 4.2 Learning

Since the *soft mask* $e_t$ is a continuous function of the model parameters, the scheduler can be learned through back-propagation. However, we have found that the naive approach of using a fixed sharpness parameter and simply minimizing the negative log-likelihood defined in Equation 1 led to the model being stuck in a local optimum which updates all dimensions at every step. We found that the following two modifications allowed the model to learn better parametrizations.

First, we can encourage $m_t$ to be either close or no greater than a target $\bar{m}$ at all time by adding a penalty term $\Omega$ to the objective. For example, we can apply a $\ell_1$ or $\ell_2$ penalty to values of $m_t$ that are greater than the target, or that simply diverge from it (in which case we also discourage the model from using too few dimensions). The cost function defined in Equation 1 then becomes:

$$\mathcal{O}(\mathbf{w}, U, V, O, u, v, b) = \mathcal{L}(\mathbf{w}, U, V, O, u, v, b) + \Omega(m, \bar{m}). \tag{10}$$

Secondly, for the model to be able to explore the effect of using fewer or more dimensions, we need to start training with a smooth mask (small $\lambda$ parameter), since for small values of $\lambda$, the model actually uses the whole hidden state. We can then gradually increase the sharpness parameter until the model truly does a partial update.

## 5 Experiments

We ran experiments with the Variable Computation variants of the Elman and Gated Recurrent Units (VCRNN and VCGRU respectively) on several sequence modeling tasks. All experiments were run using a symmetrical $\ell_1$ penalty on the scheduler $m$, that is, penalizing $m_t$ when it is greater or smaller than target $\bar{m}$, with $\bar{m}$ taking various values in the range $[0.2, 0.5]$. In all experiments, we start with a sharpness parameter $\lambda = 0.1$, and increase it by 0.1 per epoch to a maximum value of 1.

In each of our experiments, we are interested in investigating two specific aspects of our model. On the one hand, do the time patterns that emerge agree with our intuition of the time dynamics expressed in the data? On the other hand, does the Variable Computation Unit (VCU) yield a good predictive model? More specifically, does it lead to lower perplexity than a constant computation counterpart which performs as many or more operations? In order to be able to properly assess the efficiency of the model, and since we do not know *a priori* how much computation the VCU uses, we always report the "equivalent RNN" dimension (noted as RNN-d in Table 3) along with the performance on test data, *i.e.* the dimension of an Elman RNN that would have performed the same amount of computation. Note that the computational complexity gains we refer to are exclusively in terms of lowering the number of operations, which does not necessarily correlate with a speed up of training when using general purpose GPU kernels; it is however a prerequisite to achieving such a speed up with the proper implementation, motivating our effort.

We answer both of these questions on the tasks of music modeling, bit and character level language modeling on the Penn Treebank text, and character level language modeling on the Text8 data set as well as two languages from the Europarl corpus.

## 5.1 MUSIC MODELING

We downloaded a corpus of Irish traditional tunes from https://thesession.org and split them into a training validation and test of 16,000 (2.4M tokens), 1,511 (227,000 tokens) and 2,000 (288,000 tokens) melodies respectively. Each sub-set includes variations of melodies, but no melody has variations across subsets. We consider each (pitch, length) pair to be a different symbol; with rests and bar symbols, this comes to a total vocabulary of 730 symbols.

Table 1 compares the perplexity on the test set to Elman RNNs with equivalent computational costs: an VCRNN with hidden dimension 500 achieves better perplexity with fewer operations than an RNN with dimension 250.

Looking at the output of the scheduler on the validation set also reveals some interesting patterns. First, bar symbols are mostly ignored: the average value of $m_t$ on bar symbols is $0.14$, as opposed to $0.46$ on all others. This is not surprising: our pre-processing does not handle polyphony or time signatures, so bars en up having different lengths. The best thing for the model to do is then just to ignore them and focus on the melody. Similarly, the model spends lest computation on rests ($0.34$ average $m_t$), and pays less attention to repeated notes ($0.51$ average for $m_t$ on the first note of a repetition, $0.45$ on the second).

| unit type | equivalent RNN | perplexity |
|---|---|---|
| RNN-200 | – | 9.13 |
| RNN-250 | – | 8.70 |
| VCRNN-500 | 233 | 8.51 |

Table 1: Music modeling, test set perplexity on a corpus of traditional Irish tunes. Our model manages to achieve better perplexity with less computation than the Elman RNN.

We also notice that the model needs to do more computation on fast passages, which often have richer ornamentation, as illustrated in Table 2. While it is difficult to think *a priori* of all the sorts of behaviors that could be of interest, these initial results certainly show a sensible behavior of the scheduler on the music modeling task.

| note length | 0.25 | 1/3 | 0.5 | 0.75 | 1 | 1.5 | 2 |
|---|---|---|---|---|---|---|---|
| average $m$ | 0.61 | 0.77 | 0.39 | 0.59 | 0.44 | 0.46 | 0.57 |

Table 2: Average amount of computation ($m_t$) for various note lengths. More effort is required for the faster passages with 16th notes and triplets.

## 5.2 BIT AND CHARACTER LEVEL LANGUAGE MODELING

We also chose to apply our model to the tasks of bit level and character level language modeling. Those appeared as good applications since we know *a priori* what kind of temporal structure to look for: ASCII encoding means that we expect a significant change (change of character) every 8 bits in bit level modeling, and we believe the structure of word units to be useful when modeling text at the character level.

### 5.2.1 PENN TREEBANK AND TEXT8

We first ran experiments on two English language modeling tasks, using the Penn TreeBank and Text8 data sets. We chose the former as it is a well studied corpus, and one of the few corpora for which people have reported bit-level language modeling results. It is however quite small for our purposes, with under 6M characters, which motivated us to apply our models to the larger Text8 data set (100M characters). Table 3 shows bit per bit and bit per character results for bit and character level language modeling. We compare our results with those obtained with standard Elman RNN, GRU, and LSTM networks, as well as with the Conditional RNN of (Bojanowski et al., 2015).

| **Bit level PTB** | | |
|---|---|---|
| unit type | RNN-d | bpb |
| RNN-100 | 100 | 0.287 |
| RNN-500 | 500 | 0.227 |
| RNN-1000 | 1000 | 0.223 |
| CRNN-100 | 140 | 0.222 |
| VCRNN-1000 | 340 | 0.231 |
| VCRNN-1000 | 460 | **0.215** |

| **Character level PTB** | | |
|---|---|---|
| unit type | RNN-d | bpc |
| GRU-1024 | 1450 | **1.42** |
| LSTM-1024 | 2048 | **1.42** |
| RNN-1024 | 1024 | 1.47 |
| CRNN-500 | 700 | 1.46 |
| VCRNN-1024 | 760 | 1.46 |
| RNN-760 | 760 | 1.47 |
| LSTM-380 | 760 | 1.44 |
| GRU-538 | 760 | 1.43 |
| VCGRU-1024 | 648 | **1.42** |
| LSTM-324 | 648 | 1.46 |
| GRU-458 | 648 | 1.47 |

| **Character level Text8** | | | |
|---|---|---|---|
| unit type | $\bar{m}$ | RNN-d | bpc |
| RNN-512* | - | 512 | 1.80 |
| RNN-1024* | - | 1024 | 1.69 |
| LSTM-512* | - | 1024 | 1.65 |
| LSTM-1024* | - | 2048 | 1.52 |
| RNN-512 | - | 512 | 1.80 |
| GRU-512 | - | 725 | 1.69 |
| GRU-1024 | - | 1450 | 1.58 |
| VCGRU-1024 | 0.3 | 464 | 1.69 |
| VCGRU-1024 | 0.4 | 648 | 1.64 |
| VCGRU-1024 | 0.5 | 820 | 1.63 |

Table 3: **Left:** Bits per character for character level language modeling on Penn TreeBank. CRNN refers to the Conditional RNN from (Bojanowski et al., 2015). **Middle:** Bits per bit for bit level language modeling on Penn TreeBank. **Right:** Bits per character for character level language modeling on Text8. *From (Zhang et al., 2016)

**Quantitative Results.** We first compare the VCRNN to the regular Elman RNN, as well as to the Conditional RNN of (Bojanowski et al., 2015), which combines two layers running at bit and character level for bit level modeling, or character and word level for character level modeling. For bit level language modeling, the VCRNN not only performs fewer operations than the standard unit, it also achieves better performance. For character level modeling, the Elman model using a hidden dimension of 1024 achieved 1.47 bits per character, while our best performing VCRNN does slightly better while only requiring as much computation as a dimension 760 Elman unit. While we do slightly more computation than the Conditional RNN, it should be noted that our model is not explicitly given word-level information: it learns how to summarize it from character-level input.

The comparison between the constant computation and Variable Computation GRU (VCGRU) follows the same pattern, both on the PTB and Text8 corpora. On PTB, the VCGRU with the best validation perplexity performs as well as a GRU (and LSTM) of the same dimension with less than half the number of operations. On Text8, the VCGRU models with various values of the target $\bar{m}$ always achieve better perplexity than other models performing similar or greater numbers of operations. It should be noted that none of the models we ran on Text8 overfits significantly (the training and validation perplexities are the same), which would indicate that the gain is not solely a matter of regularization.

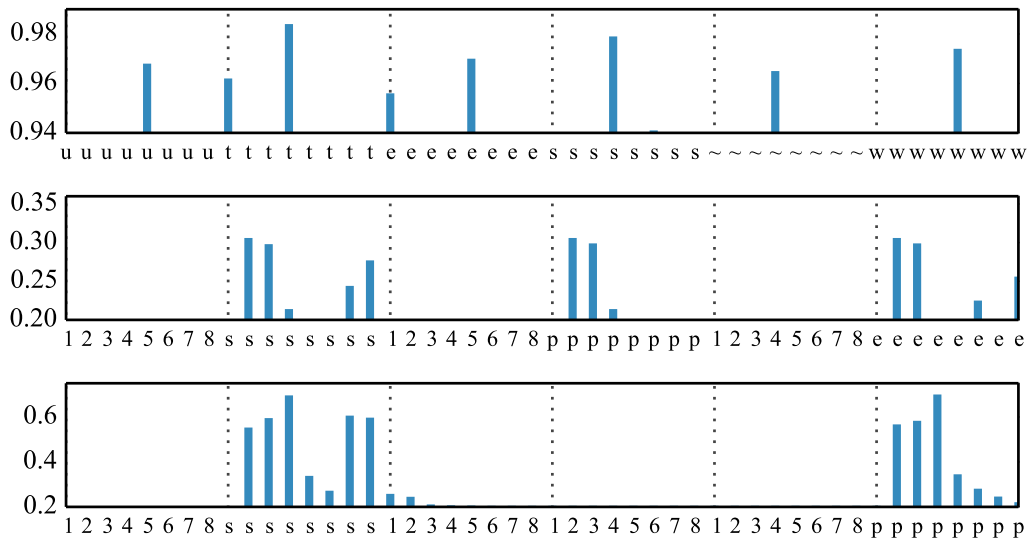

Figure 2: **Top:** Per-bit computation by VCRNN, higher dimensions (950 to 1000). **Middle**: adding 8 bits of buffer between every character. **Bottom**: adding 24 bits of buffer between each character.

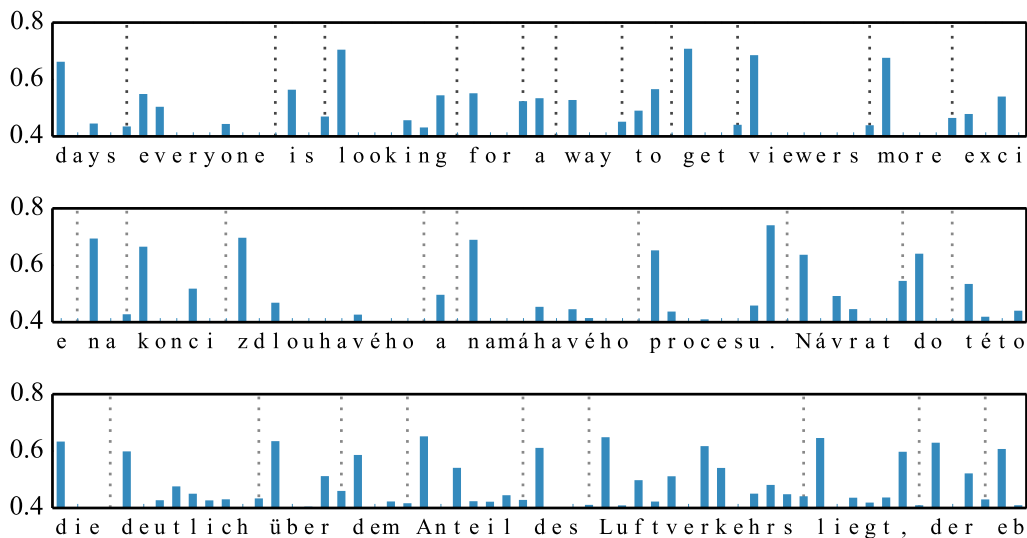

Figure 3: Per-character computation by VCRNN. Top: English. Middle: Czech. Bottom: German. All languages learn to make use of word units.

**Bit Level Scheduler.** The scheduler in the bit level language model manages to learn the structure of ASCII encoding: Figure 2 shows that the higher dimensions are modified roughly every 8 bits. We also created some artificial data by taking the PTB text and adding 8 or 24 0 bits between each character. Figure 2, shows that the model learns to mostly ignore these "buffers", doing most of its computation on actual characters.

**Character Level Scheduler.** On character level language modeling, the scheduler learns to make use of word boundaries and some language structures. Figure 3 shows that the higher dimensions are used about once per words, and in some cases, we even observe a spike at the end of each morpheme (long-stand-ing, as shown in Figure 5). While we provide results for the VCRNN specifically in this Section, the VCGRU scheduler follows the same patterns.

### 5.2.2 EUROPARL CZECH AND GERMAN

We also ran our model on two languages form the Europarl corpus. We chose Czech, which has a larger alphabet than other languages in the corpus, and German , which is a language that features long composite words without white spaces to indicate a new unit. Both are made up of about 20M characters. We tried two settings. In the "guide" setting, we use the penalty on $m_t$ to encourage the model to use more dimensions on white spaces. The "learn" setting is fully unsupervised, and encourages lower values of $m_t$ across the board.

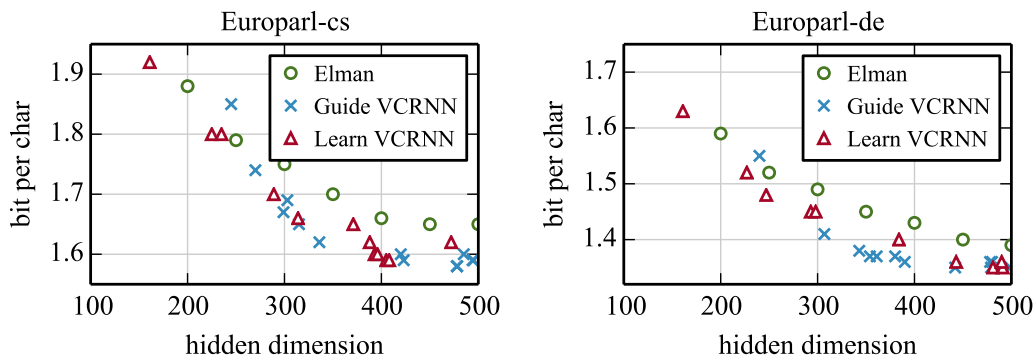

Figure 4: Bits per character for different computational loads on the Europarl Czech (left) and German (right) datasets. The VCRNN, whether guided to use boundaries or fully unsupervised, achieves better held-out log-likelihood more efficiently than the standard RNN.

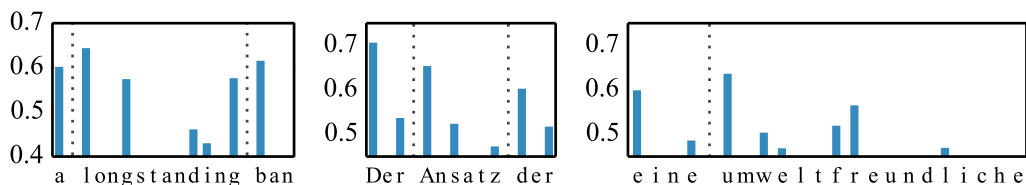

Figure 5: Per-character computation by VCRNN. The model appears to make use of morphology, separating sub-word units.

Figure 4 shows that both perform similarly on the Czech dataset, achieving better performance more efficiently than the standard RNN. On German, the guided settings remains slightly more efficient than the fully learned one, but both are more efficient than the RNN and achieve the same performance when using more dimensions. Both learn to use more dimensions at word boundaries as shown in Figure 3. The German model also appears to be learning interesting morphology (Luft-ver-kehrs, eben-falls in Figure 3, An-satz, Um-welt-freund-lich in Figure 5), and grammar (focusing on case markers at the end of articles, Figure 5).

## 6 CONCLUSION AND FUTURE WORK

In this work, we have presented two kinds of Variable Computation recurrent units: the VCRNN and VCGRU, which modify the Elman and Gated Recurrent Unit respectively to allow the models to achieve better performance with fewer operations, and can be shown to find time patterns of interest in sequential data. We hope that these encouraging results will open up paths for further exploration of adaptive computation paradigms in neural networks in general, which could lead to more computation-efficient models, better able to deal with varying information flow or multi-scale processes. We also see a few immediate possibilities for extensions of this specific model. For example, the same idea of adaptive computation can similarly be applied to yet other commonly used recurrent units, such as LSTMs, or to work within the different layers of a stacked architecture, and we are working on adapting our implementation to those settings. We also hope to investigate the benefits of using stronger supervision signals to train the scheduler, such as the entropy of the prediction, to hopefully push our current results even further.

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

# A  APPENDIX

## A.1  NEW UNITS: VCRNN AND VCGRU

We apply the method outlined in the previous paragraph to two commonly used architecture. Recall that, given a proportion of dimensions to use $m_t \in [0, 1]$ and a sharpness parameter $\lambda$, we the gating vector $e_t \in \mathbb{R}^D$ is defined as:

$$\forall i \in 1, \dots, D, \ (e_t)_i = \text{Thres}_\epsilon \left( \sigma \big( \lambda (m_t D - i) \big) \right). \tag{11}$$

The masked versions of the previous hidden and current input vectors respectively are then:

$$\bar{h}_{t-1} = e_t \odot h_{t-1} \quad \text{and} \quad \bar{i}_t = e_t \odot x_t. \tag{12}$$

First, we derive a variable computation version of the Elman RNN to get the Variable Computation Recurrent Neural Network (VCRNN) by transforming Equation 3 as follows:

$$h_t = e_t \odot g(\bar{h}_{t-1}, \bar{x}_t) + (1 - e_t) \odot h_t. \tag{13}$$

Secondly, we obtain the Variable Computation Gated Recurrent Unit (VCGRU) by deriving the variable computation of the GRU architecture. This is achieved by modifying Equations 4 to 6 as follows:

$$r_t = \sigma(U_r \bar{h}_{t-1} + V_r \bar{x}_t), \qquad z_t = e_t \odot \sigma(U_z \bar{h}_{t-1} + V_z \bar{x}_t) \tag{14}$$

$$\tilde{h}_t = \tanh(U(r_t \odot \bar{h}_{t-1}) + V \bar{x}_t) \tag{15}$$

And:

$$h_t = z_t \odot \tilde{h}_t + (1 - z_t) \odot h_{t-1} \tag{16}$$

