# Peer review of "Variable Computation in Recurrent Neural Networks"

_ICLR 2017 — accepted_

[Official Review · AnonReviewer3 · rating 4 · confidence 5 · 16 Dec 2016]

TLDR: The authors present Variable Computation in Recurrent Neural Networks (VCRNN). VCRNN is similar in nature to Adaptive Computation Time (Graves et al., 2016). Imagine a vanilla RNN, at each timestep only a subset (i.e., "variable computation") of the state is updated. Experimental results are not convincing, there is limited comparison to other cited work and basic LSTM baseline.

=== Gating Mechanism ===
At each timestep, VCRNN generates a m_t vector which can be seen as a gating mechanism.  Based off this m_t vector, a D-first (D-first as in literally the first D RNN states) subset of the vanilla RNN state is gated to be updated or not. Extra hyperparams epsilon and \bar{m} are needed -- authors did not give us a value or explain how this was selected or how sensitive and critical these hyperparms are.

This mechanism while novel, feels a bit clunky and awkward. It does not feel well principled that only the D-first states get updated, rather than a generalized solution where any subset of the state can be updated.

A short section in the text comparing to the soft-gating mechanisms of GRUs/LSTMs/Multiplicative RNNs (Wu et al., 2016) would be nice as well.

=== Variable Computation ===
One of the arguments made is that their VCRNN model can save computation versus vanilla RNNs. While this may be technically true, in practice this is probably not the case. The size of the RNNs they compare to do not saturate any modern GPU cores. In theory computation might be saved, but in practice there will probably be no difference in wallclock time. The authors also did not report any wallclock numbers, which makes this argument hard to sell.

=== Evaluation ===
This reviewer wished there was more citations to other work for comparison and a stronger baseline (than just a vanilla RNN). First, LSTMs are very simple and quite standard nowadays -- there is a lack of comparison to any basic stacked LSTM architecture in all the experiments.

The PTB BPC numbers are quite discouraging as well (compared to state-of-the-art). The VCRNN does not beat the basic vanilla RNN baseline. The authors also only cite/compare to a basic RNN architecture, however there has been many contributions since a basic RNN architecture that performs vastly better. Please see Chung et al., 2016 Table 1. Chung et al., 2016 also experimented w/ PTB BPC and they cite and compare to a large number of other (important) contributions.

One cool experiment the authors did is graph the per-character computation of VCRNN (i.e., see Figure 2). It shows after a space/word boundary, we use more computation! Cool! However, this makes me wonder what a GRU/LSTM does as well? What is the magnitude of the of the change in the state vector after a space in GRU/LSTM -- I suspect them to do something similar.

=== Minor ===
* Please add Equations numbers to the paper, hard to refer to in a review and discussion!

References
Chung et al., "Hierarchical Multiscale Recurrent Neural Networks," in 2016.
Graves et al., "Adaptive Computation Time for Recurrent Neural Networks," in 2016.
Wu et al., "On Multiplicative Integration with Recurrent Neural Networks," in 2016.

[Official Review · AnonReviewer4 · rating 7 · confidence 4 · 16 Dec 2016 (modified: 20 Jan 2017)]
**Interesting exploratory work.**

This is high novelty work, and an enjoyable read.

My concerns about the paper more or less mirror my pre-review questions. I certainly agree that the learned variable computation mechanism is obviously doing something interesting. The empirical results really need to be grounded with respect to the state of the art, and LSTMs are still an elephant in the room. (Note that I do not consider beating LSTMs, GRUs, or any method in particular as a prerequisite for acceptance, but the comparison nevertheless should be made.)

In pre-review responses the authors brought up that LSTMs perform more computation per timestep than Elman networks, and while that is true, this is an axis along which they can be compared, this factor controlled for (at least in expectation, by varying the number of LSTM cells), etc. A brief discussion of the proposed gating mechanism in light of the currently popular ones would strengthen the presentation.

---
2017/1/20: In light of my concerns being addressed I'm modifying my review to a 7, with the understanding that the manuscript will be amended to include the new comparisons posted as a comment.

[Official Review · AnonReviewer2 · rating 7 · confidence 4 · 16 Dec 2016 (modified: 27 Jan 2017)]
**Review: Variable Computation in Recurrent Neural Networks**

This paper describes a simple but clever method for allowing variable amounts of computation at each time step in RNNs. The new architecture seems to outperform vanilla RNNs on various sequence modelling tasks. Visualizations of the assignment of computational resources over time support the hypothesis that the model is able to learn to assign more computations whenever longer longer term dependencies need to be taken into account.

The proposed model is evaluated on a multitude of tasks and its ability to outperform similar architectures seems consistent. Some of the tasks allow for an interesting analysis of the amount of computation the model requests at each time step. It’s very interesting to see how the model seems to use more resources at the start of each word or ASCII character. I also like the investigation of the effect of imposing a pattern of computational budget assignment which uses prior knowledge about the task. The superior performance of the architecture is impressive but I’m not yet convinced that the baseline models had an equal number of hyperparameters to tune. I’ll come back to this point in the next paragraph because it’s mainly a clarity issue.

The abstract claims that the model is computationally more efficient than regular RNNs. There are no wall time measurements supporting this claim. While the model is theoretically able to save computations, the points made by the paper are clearly more conceptual and about the ability of the model to choose how to allocate its resources. This makes the paper interesting enough by itself but the claims of computational gains are misleading without actual results to back them up. I also find it unfortunate that it’s not clear from the text how the hyperparameter \bar{m} was chosen. Whether it was chosen randomly or set using a hyperparameter search on held-out data influences the fairness of a comparison with RNNs which did not have a similar type of hyperparameter for controlling regularization like for example dropout or weight noise (even if regularization of RNNs is a bit tricky). I don’t consider this a very serious flaw because I’m impressed enough by the fact that the new architecture achieves roughly similar performance while learning to allocate resources but I do think that details of this type are too important to be absent from the text. Even if the superior performance is due to this extra regularization controlling parameter it can actually be seen as a useful part of the architecture but it would be nice to know how sensitive the model is to its precise value.

To my knowledge, the proposed architecture is novel. The way the amount of computation is determined is unlike other methods for variable computation I have seen and quite inventive. Originality is one of this paper’s strongest points. 

It’s currently hard to predict whether this method for variable computation will be used a lot in practice given that this also depends on how feasible it is to obtain actual computational gains at the hardware level. That said, the architecture may turn out to be useful for learning long-term dependencies. I also think that the interpretability of the value m_t is a nice property of the method and that it’s visualizations are very interesting. It might shed some more light into what makes certain tasks difficult for RNNs. 

Pros:
Original clever idea.
Nice interesting visualizations.
Interesting experiments.

Cons:
Some experimental details are not clear.
I’m not convinced of the strength of the baseline.
The paper shouldn’t claim actual computational savings without reporting wall-clock times.

Edit:
I'm very positively impressed by the way the authors ended up addressing the biggest concerns I had about the paper and raised my score. Adding an LSTM baseline and results with a GRU version of the model significantly improves the empirical quality of the paper. On top of that, the authors addressed my question about some experimental detail I found important and promised to change the wording of the paper to remove confusion about whether the computational savings are conceptual or in actual wall time. I think it's fine that they are conceptual only as long as this is clear from the paper and abstract. I want to make clear to the AC that since the changes to the paper are currently still promises, my new score should be assumed to apply to an updated version of the paper in which the aforementioned concerns have indeed been addressed. 

Edit: 
Since I didn't know that the difference with the SOTA for some of these tasks was so large, I had to lower my score again after learning about this. I still think it's a good paper but with these results I cannot say that it stands out.

[Author Response · Yacine Jernite · 14 Jan 2017]
**To Reviewers 2, 3 and 4:**

We thank the reviewers for several helpful suggestions. We first address the issues raised by two or more reviewers here. As a general comment, we would like to emphasize again the exploratory nature of our work. We aimed to explore the following question: is it possible for a recurrent model to (cheaply) determine a priori  how much computation it is going to need at each time step? We designed our model and experimental section with this interrogation in mind, and believe that we were able to provide a positive answer.

As regards the wall-clock time, the computational gains are currently, to quote Reviewer 2 conceptual only, as we are using the Torch GPU kernels, and optimizing the hardware implementation was beyond the scope of this project. Our goal was to make the possible implications of the result clear, but we will clarify the language of the paper to reflect this more accurately.

For each experiment, we tried a range of \bar{m} values for the Variable Computation recurrent network and a range of hidden state dimensions for its vanilla counterpart. While we only provided one result on PTB, Figure 5 illustrates how both of these choices affect the model’s computation and perplexity on a Czech and a German dataset; note that the Variable Computation curve for perplexity/number of operations is always under the Vanilla one, meaning fewer operations for a better perplexity. We will provide the same figure for PTB.

Our original reasoning for only providing vanilla RNN results on PTB was that we were trying to compare apples to apples, so to speak, and to reinforce the message that we are considering a paradigm which can be extended to other architectures.

However, we have since successfully applied our method to the GRU architecture, which gives us the Variable Computation Gated Recurrent Unit (VCGRU). Between this and all of the reviewers’ well-received advice to provide more context, we re-ran experiments with the RNN and VCRNN, GRU and VCGRU, and the LSTM architecture. The results are as follow. As in the paper, the second column provides the dimension of a vanilla RNN which would perform the same number of operations.

------------------------------------------------------------------------
Model			| RNN-equiv	| train / valid / test
------------------------------------------------------------------------
LSTM_1024		| 2048		| 1.21 / 1.46 / 1.42
GRU_1024		| 1450		| 1.17 / 1.47 / 1.42
------------------------------------------------------------------------
RNN_1024		| 1024		| 1.31 / 1.51 / 1.47
LSTM_512		| 1024		| 1.31 / 1.49 / 1.44
GRU_725		| 1024		| 1.26 / 1.47 / 1.43
------------------------------------------------------------------------
VCRNN_1024		| 760 		| 1.39 / 1.51 / 1.46
RNN 760		        | 760 		| 1.37 / 1.52 / 1.47
LSTM 380 		| 760 		| 1.34 / 1.48 / 1.43
GRU 538		        | 760 		| 1.28 / 1.49 / 1.44
------------------------------------------------------------------------
VCGRU_1024		| 648 		| 1.30 / 1.47 / 1.42
LSTM 324 		| 648 		| 1.41 / 1.51 / 1.46
GRU 458		        | 648 		| 1.37 / 1.52 / 1.47
------------------------------------------------------------------------
VCGRU_1024		| 538 		| 1.32 / 1.49 / 1.44
------------------------------------------------------------------------

The VCGRU behaves similarly to the VCRNN: the number of operations per character follows the same patterns, and it achieves the same perplexity as its constant computation counterpart with fewer operations. It also outperforms both GRUs and LSTMs for a given number of operations. We will add these results and discussion in the paper.

Finally, we will add further discussion of our gating mechanism and comparison to other existing ones in the next version of the paper. Compared to the rest of the literature, the VCRNN gating paradigm is closest to that of the GRU: the main difference between the two is that where the GRU gate can take any shape, ours is more constrained and parametrized by a single value m_t, which allows to cheaply control the number of operations for the current time step.

We address the other comments in individual replies.

[Author Response · Yacine Jernite · 20 Jan 2017]
**New Version**

We would like to thank the reviewers again for a discussion which we feel helped us improve the paper significantly. We have provided a newer version which implements the items we discussed in response to their initial comments. Most notably:
- We added the description of the Variable Computation GRU to the paper
- We added comparisons to the LSTM and GRU architectures on the Penn TreeBank dataset
- We added experiments on the larger English corpus Text8 (

[Final Decision · Program Chairs · 06 Feb 2017]
**ICLR committee final decision**

This paper describes a new way of variable computation, which uses a different number of units depending on the input. This is different from other methods for variable computation that compute over multiple time steps. The idea is clearly presented and the results are shown on LSTMs and GRUs for language modeling and music prediction.
 
 Pros:
 - new idea
 - convincing results in a head to head comparison between different set ups.
 
 Cons:
 - results are not nearly as good as the state of the art on the reported tasks.
 
 The reviewers and I had several rounds of discussion on whether or not to accept the paper. One reviewer had significant reservations about the paper since the results were far from SOTA. However, since getting SOTA often requires a combination of several tricks, I felt that perhaps it would not be fair to require this, and gave them the benefit of doubt (especially because the other two reviewers did not think this was a dealbreaker). In my opinion, the authors did a fair enough job on the head to head comparison between their proposed VCRNN models and the underlying LSTMs and GRUs, which showed that the model did well enough.